# The Impact of a Cypovirus on Parental and Filial Generations of *Lymantria dispar* L.

**DOI:** 10.3390/insects14120917

**Published:** 2023-11-30

**Authors:** Yuriy B. Akhanaev, Sergey V. Pavlushin, Daria D. Kharlamova, Daria Odnoprienko, Anna O. Subbotina, Irina A. Belousova, Anastasia N. Ignatieva, Anastasia G. Kononchuk, Yuri S. Tokarev, Vyacheslav V. Martemyanov

**Affiliations:** 1Institute of Systematics and Ecology of Animals, SB RAS, Frunze Str. 11, Novosibirsk 630091, Russia; 2Institute of Biology, Irkutsk State University, Karl Marx Str. 1, Irkutsk 664003, Russia; 3Department of Molecular Biology and Biotechnology, Novosibirsk State University, Pirogova Str. 1, Novosibirsk 630090, Russia; 4All-Russian Institute of Plant Protection, Sch. Podbelskogo 3, Pushkin, St. Petersburg 196608, Russia

**Keywords:** spongy moth, *Lymantria dispar*, cypovirus, microsporidia, virulence

## Abstract

**Simple Summary:**

The spongy moth *Lymantria dispar* L., formerly known as the gypsy moth, is an important forest pest in the Holarctic region. To control this pest, the use of microbial pathogens is important because they are usually possess more selective and environmentally safe than chemical insecticides. In this study, we estimated the virulence of a new strain of cypovirus-1 isolated from the Siberian moth *Dendrolimus sibiricus* (DsCPV-1) against *L. dispar* larvae and its impact on a subsequent *L. dispar* generation. We observed increased insect mortality and longer larval development times following DsCPV-1 infection. DsCPV-1 did not affect the fecundity of surviving females, but considerably reduced the hatching ability of eggs of these females. Offspring of surviving parents that had been challenged with DsCPV-1 were tested to see if the virus affected their susceptibility to a stress factor (starvation). In addition, we checked the ability of cypovirus-1 to be transmitted vertically via an alternative host (spongy moth). Estimation of this feature may be useful for microbial control. Although the DsCPV-1 infection of parents had adverse effects on the spongy moth larvae, DsCPV-1 was not detectable in their offspring.

**Abstract:**

Recently, we found that the spongy moth *Lymantria dispar* L. is susceptible to infection by a Dendrolimus sibiricus cytoplasmic polyhedrosis virus (DsCPV-1). In the present study, we evaluated the pathogenicity of DsCPV-1 against *L. dispar* larvae and its impact on surviving insects after the infection. Offspring of virally challenged insects were tested for susceptibility to a stress factor (starvation). In addition, we used light microscopy and quantitative polymerase chain reaction (qPCR) to test the ability of DsCPV-1 to be transmitted vertically. We found insect mortality of the *L. dispar* parents following the infection was positively associated with DsCPV-1 dose. DsCPV-1 was lethal to second-instar *L. dispar* larvae with a 50% lethal dose (LD_50_) of 1687 occlusion bodies per larva. No vertical transmission of DsCPV-1 to offspring larvae was detected, while the majority of insect deaths among offspring larvae were caused by microsporidia (*Vairimorpha lymantriae*), which was harbored by the parents. The offspring of virally challenged parents exhibited a higher number of detected microsporidia compared to the control. Our findings suggest that the application of DsCPV-1 is effective in controlling pests in terms of transgenerational impact following virus exposure.

## 1. Introduction

Entomopathogenic microorganisms, including viruses, fungi, bacteria, microsporidia, and others, have substantial potential for controlling insect pests. Their natural occurrence affects a host’s population density, owing to their virulence and the ability to be transmitted horizontally or vertically. The use of these microorganisms as biological control agents against pest species is considered an alternative to chemical insecticides [1]. Some of them have been successfully released for insect pest suppression [2]. Therefore, for microbial control, it remains an important goal to continuously explore isolates and candidate species with a suitable host range, appropriate environmental tolerance, greater virulence, and other beneficial traits [3].

Among entomopathogenic microorganisms, viruses are a popular entomopathogen used for pest control [2]. A considerable proportion of studied viruses are representatives of the Baculoviridae family (nucleopolyhedroviruses [NPVs]) [1]. Baculoviruses have a double-stranded circular DNA genome that can produce protein structures known as occlusion bodies (OBs), which allow them to persist outside their hosts for a long time [4]. OBs usually have a typical polyhedral shape and large size, and are easily detectable via light microscopy. Cytoplasmic polyhedrosis viruses or cypoviruses (CPVs; the Spinareoviridae family [5]) are characterized by a double-stranded linear segmented RNA genome [6], and have been studied to a lesser extent than baculoviruses. Just as NPVs, CPVs produce polyhedral OBs that are not homologous; however, some CPV isolates frequently produce cubic OBs or OBs with angular sides [7,8,9,10]. In general, cypoviruses infect midgut epithelial cells of insect larvae, and cause chronic rather than lethal diseases.

NPVs and CPVs both use horizontal and vertical routes of transmission within host populations [9,11]. Horizontal transmission enables parasites (or pathogens) to infect susceptible hosts and disseminate spatially. As for vertical transmission, it ensures a parasite’s (or pathogen’s) transfer from a parental generation to offspring. Vertical transmission might include transovum transmission via the egg surface and transovarian transmission within the egg or transmission of covert infections. The covert infections can be activated in virus-harboring hosts and transformed to overt infections when the hosts are stressed [12,13]. Research on parasites’ capacity for vertical transmission and for activation from covert to overt infection can be potentially useful in microbial control.

The spongy moth *Lymantria dispar* L. (Lepidoptera: Erebidae), formerly known as the gypsy moth, is a destructive forestry pest in the Holarctic region [14]. *L. dispar* populations are always under the influence of both abiotic and biotic factors, leading to the formation of population cycles [15]. Natural enemies, especially pathogenic microorganisms, are known to be an important cause of the population cycles. The role of natural enemies, especially pathogens, in the biocontrol of the spongy moth is well documented [16]. *L. dispar* multiple nucleopolyhedrovirus (LdMNPV) is a dominant infectious agent contributing to the regulation of outbreak populations [17], and has served as the basis of biological insecticides for spongy moth control [16].

In a recent study, we isolated and investigated a new cypovirus from naturally infected *Dendrolimus sibiricus* larvae (DsCPV-1) [18]. In the present work, we studied DsCPV-1 against *L. dispar* larvae, and then estimated the impact on surviving parents and an indirect impact on a filial generation of *L. dispar*. We also estimated the capacity of DsCPV-1 for vertical transmission via an alternative host and the tolerance of the filial *L. dispar* generation to a stress factor.

## 2. Materials and Methods

### 2.1. Insects and DsCPV-1

*L. dispar* egg masses were collected in the Ongudai district of Altai Republic, Russia (50.75° N, 86.10° E). The egg surface was decontaminated with 6% hydrogen peroxide for 10 min followed by rinsing in sterile water. Then, the eggs were placed into Petri dishes at 100 eggs per dish and kept at a constant temperature (28 °C) for 48 h to induce egg hatching. Next, neonate *L. dispar* larvae were placed in ventilated plastic containers (approximately 100 larvae in a 20 L container) and fed leaves of cut branches of silver birch (*Betula pendula* Roth).

DsCPV-1 OBs were isolated from cadavers of *D. sibiricus* larvae. A detailed description is presented in [18]. Briefly, the *D. sibiricus* larval cadavers were homogenized in sterile water, filtered through three layers of cheesecloth, and centrifuged at 20,000× *g* for 20 min. The OB concentration was determined using a hemocytometer under phase contrast microscopy. The pelleted OBs were used for infection of *L. dispar* larvae (see below).

### 2.2. Infection of a Parental L. dispar Generation with DsCPV-1

The droplet-feeding method was used to infect *L. dispar* larvae by DsCPV-1 [19]. Each individual newly molted second-instar *L. dispar* larva was offered a droplet (0.5 µL) of 100 mg ml^−1^ sucrose with 0.01 mg mL^−1^ red dye and one of five doses of DsCPV-1. In particular, 0.5, 5, 50, 500 and 5000 OBs per larva were employed. The larvae were starved for approximately 4 h prior to inoculation. Larvae in a control group were fed with virus-free water droplets. Larvae that ingested the whole droplet were placed in a 350 mL ventilated plastic container with a host plant. The insects were reared under laboratory conditions at a constant temperature (23 °C) and with a natural daylight regime. The larvae that failed to consume the whole droplet were excluded from the bioassay. Five or four replicates were used with 10 larvae per container for each DsCPV-1 dose. Insect mortality was monitored daily 15 d after infection.

Surviving *L. dispar* larvae infected with 50 and 500 OBs of DsCPV-1 were allowed to become pupae. After pupation, they were placed into a plastic container (125 mL) with damp cotton wool. Once the adults emerged, they were moved to a new 1000 mL plastic container (1♀:1♂) with a paper substrate for mating and oviposition. The obtained egg masses were kept in a plastic container at room temperature for 2 months for postembryonic development. Next, the eggs were placed in a refrigerator at 2 °C for winter diapause from September to April. Insect mortality, postinfection larval development, fecundity of surviving females, and larvae hatching were determined in the parental generation.

### 2.3. The Impact of DsCPV-1 Infection on a Filial L. dispar Generation

Egg masses (from the surviving parents: n = 13 for control, n = 27 and 26 for 50 and 500 OBs, respectively) next spring after overwintering were transferred to 28 °C to trigger larva hatching. After that, the neonate *L. dispar* larvae were transferred to a container with the host plant; the container was labeled according to the parental generation (i.e., either control or infected parents). Newly molted fourth-instar *L. dispar* larvae were randomly selected and placed in individual plastic containers (125 mL), and each experimental line (derived from 50 and 500 OBs-infected parents including the control) was divided into two identical groups. The first group was fed continuously with leaves of the host plant until pupation (without starvation). Meanwhile, the second group of the larvae was starved for 4 days to activate possible vertically transmitted DsCPV-1. After 4 days of the starvation, the feeding was resumed. Mortality was recorded every 2 days until all insects either died or pupated. In total, 124 and 159 larvae from 50 and 500 OBs, respectively, and 75 larvae from the control were used. All dead larvae were examined via light microscopy (Axioscope 40 Carl Zeiss, Goettingen, Germany) at 1000× magnification to determine the presence of OBs of DsCPV-1.

### 2.4. qPCR for Assessing the Possible Vertical Transmission of DsCPV-1

qPCR was used to quantify potential vertically transmitted DsCPV-1 within individual larvae of offspring that died as a result of the stress factor (starvation). For this purpose, 10 samples of starved larval cadavers from the control and experimental groups (i.e., derived from 50 and 500 OBs-infected parents) were analyzed. Until further analysis, the samples were stored at −80 °C. Primers were designed by means of Primer-BLAST (NCBI). For this analysis, the polyhedrin gene was chosen as the most conserved. Primer sequences were (5′-3′) TCTCACCGAATGCTTACCCA and AGAGCGTCACCCTATCCGAA. Total RNA was isolated using the Lira-reagent (BioLabMix, Novosibirsk, Russia); 1000 μL of the reagent was added to 100 μL of larval tissue, homogenized with a polyurethane pestle, incubated for 3 h at room temperature, supplemented with 200 μL of chloroform, mixed, and centrifuged for 10 min at 10,000× *g*. Isopropanol and a 3 M sodium acetate solution were added to the RNA solution contained in the upper aqueous fraction. Then, the sample was incubated for 10 min at room temperature and centrifuged at 10,000× *g*. The pellet was washed three times with 75% ethyl alcohol and dissolved in diethyl pyrocarbonate–treated water. RNA concentration and purity were estimated with a spectrophotometer (c). Reverse transcription was performed using the RT-M-MuLV-RH kit (BioLabMix, Novosibirsk, Russia). dsRNA was denatured beforehand at 95 °C for 15 min. The reaction mixture for reverse transcription consisted of 1× RT-KCl buffer, 0.01 M dithiothreitol, 0.5 mM each dNTP, 1 μM random hexameric primers, 1 U of M-MuLV–RH reverse transcriptase, and 500 ng of RNA in a total volume of 20 μL. The reactions were carried out under the following conditions: 25 °C for 10 min, 42 °C for 60 min, and 70 °C for 10 min. The PCR assay was conducted using a CFX96 Touch™ Real-Time PCR Detection System (Bio-Rad Laboratories, Hercules, CA, USA). Each qPCR mixture consisted of 1× BioMaster HS-qPCR SYBR Blue (BioLabMix, Novosibirsk, Russia), 100 nM each primer, and 50 ng of DNA in a total volume of 25 μL. The reactions were performed in triplicate to ensure reproducibility and run in 96-well plates under the following conditions: 95 °C for 5 min, and 40 cycles of 95 °C for 15 s, and 60 °C for 1 min. Melting curves (60 °C to 95 °C) were constructed for each reaction to ensure a single specific PCR product. RNA isolated and converted into DNA via the same method from a suspension of polyhedra served as a positive control. Amplification efficiency was assessed for each primer set using 2-fold serial dilutions of DNA.

### 2.5. Microsporidia Detection and Identification

While analyzing the causes of insect mortality via light microscopy, we found many *L. dispar* cadavers dead from microsporidia in our experiment. Microsporidian spores were visualized at 1000× magnification under a light microscope. The cadavers were homogenized with a plastic pestle adapted for an Eppendorf tube and spun at 1000× *g* for 5 min to pellet the spores. After upper layers of tissue debris were removed, the spore pellet was investigated via fluorescence microscopy and a molecular genetic analysis.

For the fluorescence microscopy, the spores were resuspended in 100 µL of water, and a drop of this suspension (20 µL) was air-dried, fixed with methanol on a glass slide, and covered with 5 µM DAPI stain. After 5 min, the slides were examined under an epifluorescence-equipped Axio 10 Imager M1 microscope (Carl Zeiss, Goettingen, Germany).

For the molecular genetic analysis, the spore samples were processed for DNA extraction [20] and PCR amplification using the DreamTaq Kit (Thermo Fisher Scientific, Waltham, MA, USA) with primers 18f:1047r [21]. Amplicons of expected size of ~850 bp were gel-purified [22] and sequenced using the method [23] on an ABI Prism 3500 (Applied Biosystems, Foster city, CA, USA) capillary sequencer.

### 2.6. Statistical Analysis

For all statistical analyses, we used RStudio (ver. 2022.07.0) based on the R software (ver. 4.1.2) [24]. Insect mortality data and development time of infected larvae were analyzed by means of a generalized linear model (GLM) with a binomial (logit link function) and gamma distribution (identity link function), respectively. χ^2^ statistics were computed to check the significance of effects (*p* < 0.05). The “ecotox” package (v.1.4.4) [25] was used to calculate the LD_50_ from cumulative mortality data at 15 days after DsCPV-1 infection. The GLM (binomial distribution) was also used for the analysis of microsporidian prevalence in offspring larvae. The fecundity and egg hatching data showed a normal distribution (Shapiro–Wilk test, *p* > 0.05) and were analyzed by one-way analysis of variance (ANOVA) followed by Dunnett’s test (*p* < 0.05).

## 3. Results

### 3.1. The Pathogenicity of DsCPV-1 and Life History-Traits of the Parental L. dispar Generation

The droplet feeding of DsCPV-1 to parental *L. dispar* larvae led to increased larval mortalities relative to the control (χ^2^ = 10.37, df = 1, *p* < 0.001). A dose-dependent trend was detected (χ^2^ = 43.83, df = 1, *p* < 0.0001, Appendix A). To kill the parental larvae with a probability of 50%, 1687 calculated OBs (95% fiducial limits: 495–13,320 OBs) were required. Infected larvae exhibited reduced voracity, and dead larvae showed no signs of liquefaction (typical of NPVs).

Larval mortality rates of 17.5% and 33.5% were obtained after DsCPV-1 infection at 50 and 500 OBs, respectively (Figure 1A). Larval development time was significantly longer with DsCPV-1 infection compared to control larvae (χ^2^ = 0.63, df = 1, *p* < 0.0001). In particular, control larvae took as long as 24.7 ± 0.4 days (mean ± SE) to complete larval development starting from the second instar, compared to 30.0 ± 1.1 and 33.2 ± 1.4 days for insects infected with 50 and 500 OBs, respectively (Figure 1B, Dunnett’s test, *p* < 0.05).

The analysis of the fecundity of surviving females showed no significant effects of the DsCPV-1 infection (F_2,15_ = 0.28, *p* = 0.758). Nonetheless, the percentage of eggs hatching from the surviving females was negatively affected by DsCPV-1 infection (F_2,15_ = 6.51, *p* = 0.012), and was significantly lower by approximately 40% (42.0 ± 18.1% and 50.4 ± 6.3% eggs hatched for 50 and 500 OBs, respectively) as compared to the control (91.2 ± 2.4% eggs hatched; Dunnett’s test, *p* < 0.05).

### 3.2. The Impact of DsCPV-1 Infection on the L. dispar Filial Generation

The mortality of larvae reared without starvation reached 14.7% and 24.7% for offspring whose parents were infected with 50 and 500 OBs, respectively. The latter value was significantly different from the control level of 10.8% (Dunnett’s test, *p* = 0.047, Figure 2, without starvation). Starvation for 4 days resulted in a significant increase in insect mortality as compared to the larvae reared without starvation (χ^2^ = 160.28, df = 1, *p* < 0.0001). In particular, in the control (derived from uninfected parents) starved group, larval mortality reached 88.6%. Similarly, an increase in larval mortality was observed in groups derived from 50 and 500 OBs-infected parents: 79.3% and 83.3%, respectively (Figure 2, starvation). Nevertheless, we failed to detect any OBs via light microscopy among the *L. dispar* cadavers. Moreover, in those offspring larvae that died of starvation, DsCPV-1 was not detectable via qPCR. Analyzed samples of larval cadavers from both treatment groups (i.e., where parents were infected with 50 and 500 OBs) tested negative for the presence of DsCPV-1. Further analysis of etiology of the insect mortality revealed that some larval deaths were due the presence of another pathogen. Unexpectedly, we found a high prevalence of microsporidia in larval cadavers of the *L. dispar* offspring. Microsporidia were found in most larval cadavers (Figure 2). It is noteworthy that a higher mean level of detected microsporidia coincided with a higher DsCPV-1 dose applied to the parental generation (Figure 2). It should also be noted that at 500 OBs in parents, in both experimental lines (without starvation and starvation), the microsporidian infection level was significantly different from the control (Dunnett’s test, *p* < 0.001, Figure 2). Starvation for 4 days in this experiment was probably critical and caused high insect mortality due to stress of the starved larvae including the control. Therefore, we could not determine the exact reasons for the mortality in other larval cadavers.

### 3.3. Microsporidia Detection

First microscopic surveys of insect cadavers on smears of inner tissues showed the presence of multiple bodies with an appearance typical of infection with microsporidia at the late stage of the disease, when multiple spores are produced by the parasite. The spores were elongated and oval-shaped, measuring 4.14 to 5.56 (average 4.84) µm long and 1.90 to 2.57 (average 2.20) µm wide. A thick spore wall with a smooth outline was seen under the light microscope. DAPI staining revealed clearly distinguishable pairs of nuclei in a substantial proportion of the spores (Figure 3A).

The subsequent analysis included PCR-based detection with primers specific for microsporidian SSU rRNA. Positive results were obtained and a consensus sequence 1393 bp long was 100% identical to the homologous fragment of the SSU rRNA gene of *Vairimorpha (Nosema) lymantriae* (GenBank accession # AF033315), while other entries of *Vairimorpha* spp. showed lower similarity indices. To exclude horizontal contamination by *V. lymantriae*, we also analyzed a portion of eggs from the batch employed for continuation of the *L. dispar* offspring. The eggs were crushed and examined under the light microscope; multiple spores of the same size and shape were detected, just as in late-instar larvae. The presence of *V. lymantriae* spores was confirmed via PCR amplification and sequencing of the egg DNA samples. Thus, the filial generation of *L. dispar* was initially infected by microsporidia from the parental generation rather than the result of horizontal contamination.

## 4. Discussion

DsCPV-1 is a potential biocontrol agent for several lepidopteran species [18]. The results of the current study provide valuable insights into the virulence of DsCPV-1 against *L. dispar* larvae and its direct and indirect impact on both the infected and subsequent generation of *L. dispar*. A dose-dependent response to DsCPV-1 infection was observed in second-instar *L. dispar* larvae, with a calculated LD_50_ of 1687 OBs per larva. DsCPV-1 infection not only caused insect mortality, but also had an adverse effect on fitness of the parental *L. dispar* individuals that survived the infection. In addition to mortality, larva development time was found to be prolonged, which is typical for lepidopteran species following viral infection [26]. No reduction in fecundity in surviving females was observed. Possibly, the prolonged larval stage contributed to recovery of the infected larvae. Nevertheless, the DsCPV-1 infection had consequences: reduced fertility was observed in the surviving females. These features of exposure of *L. dispar* larvae to DsCPV-1 at lethal and sublethal doses should be taken into consideration in biocontrol.

Our findings indicate that DsCPV-1 was not transmitted vertically to the next *L. dispar* generation from experimentally infected parents. We failed to detect any OBs using either light microscopy or qPCR. In addition, we did not detect any structures resembling polyhedral bodies that can be attributed to a baculovirus infection, although LdMNPV is a common pathogen in *L. dispar* populations [8,17]. It should be noted the primers used in this study were unable to cross-link to the cDNA of the baculovirus gene, as the nucleotide sequences of the polyhedrin for NPV and CPV were not closely related [27,28]. Some researchers have reported that a covert viral infection can be activated when virus-harboring insects are inoculated with either NPV or CPV [29,30,31]. The presence of covert LdMNPV infection could lead to its activation when parental *L. dispar* larvae are subjected to stress, such as inoculation with DsCPV-1. Infection of the parental *L. dispar* larvae with DsCPV-1 resulted in increased insect mortality. Nonetheless, this mortality was due to larval susceptibility to DsCPV-1 and was not associated with the activation of the covert LdMNPV, as judged by the symptoms of the infected larvae (gut disintegration typical of CPV; there was no liquefaction typical of NPV infections due to chitinase activity). Earlier, Yang and coauthors [32] reported that activation of covert baculovirus infections in *L. dispar* larvae does not occur after their exposure to *Dendrolimus kikuchi* NPV. We can theorize that either the parental *L. dispar* generation was LdMNPV-free, or that the DsCPV-1 infection did not lead to activation of covert baculovirus infection.

One of possible reasons for the high microsporidian infection level is contamination of the host plant by spores and subsequent infection during the rearing of the experimental insects in the laboratory. Microsporidian infections in *L. dispar* populations have been reported previously [33,34,35]. These articles indicate that microsporidia are largely detectable in European *L. dispar* populations. The observed microsporidian infection level is rather unusual for *L. dispar* in Western Siberia, i.e., in the Asian part of *L. dispar*’s geographic range [36]. Microsporidia were not detected in recent surveys of *L. dispar* in Western Siberia [37] and this prevalent microsporidian infection was quite unexpected. According to our observations, the most prevalent pathogen in Siberian *L. dispar* populations is LdMNPV [36]. Moreover, the low probability of exogenous contamination by microsporidia is supported by the absence of high *L. dispar* population density, at least during the last 50 years on the territory where birch branches have been cut for insect feed. In other words, the risk of contamination with specific *L. dispar* microsporidia from outside is extremely low. Finally, our analysis of a portion of eggs from infected parents clearly indicated the presence of *V. lymantriae* in the *L. dispar* population under study. Vertical transmission is a common feature of microsporidia, and has been fairly well documented for *V. lymantriae* in *L. dispar* [38]. Thus, we can rule out an exogenous source of contamination of the host plant by microsporidia during the experiment.

External disinfection of eggs has been shown to reduce virus prevalence in many insect species [11]. This procedure prevents virus transmission to the next generation of the host via the transovum route. We routinely conduct decontamination of the eggs’ surface prior to insect hatching and rearing in the laboratory. In the current work, the surface of parental *L. dispar* egg masses was decontaminated, while eggs of the subsequent offspring were not decontaminated to avoid altering the probability of DsCPV-1 vertical transmission. As a result, we detected high levels of microsporidia in cadavers of *L. dispar* offspring. The detected microsporidia caused high mortality of *L. dispar* offspring, suggesting that the *L. dispar* population from the Altai Republic (used in this study) was initially infected with the microsporidia. We concluded that the detected microsporidia were transmitted to the *L. dispar* offspring via the transovarian route.

We have demonstrated that the impact of viral infection on the offspring depends on how severely the parents were infected with the virus. Specifically, microsporidia-induced mortality of normally feeding larvae increased as a function of the DsCPV-1 dose used to infect the parental generation. In starved larvae, this effect was more pronounced, and the proportion of microsporidia-induced mortality increased by a factor of two. Therefore, both factors—pre-infection of the parents with DsCPV-1 and/or starvation of the population with high microsporidian prevalence—lead to activation of chronic microsporidian infection in acute form. Bauer and coauthors [39] have shown that the interaction between LdMNPV and a microsporidium (*Nosema* sp.) during coinfection varies, but efficacy of LdMNPV is significantly increased when it is applied to *L. dispar* larvae previously infected with *Nosema* sp. either orally or transovarially. In this study, we cannot compare the efficacy of the virus between populations with different prevalence rates of microsporidia, because we examined only one population with one prevalence rate of microsporidia. Nevertheless, we clearly demonstrated the opposite: pre-infection with DsCPV-1 elevates the effectiveness of vertically transmitted microsporidia. Which types of interactions between cypoviruses and microsporidia are possible remain unclear. Further research is needed to identify the mechanisms of this interaction.

It is important to discuss the high overall insect mortality of the filial generation under the influence of starvation (Figure 2). We chose this period based on our previous study, showing that a Siberian *L. dispar* population can starve for 4 or even 5 days without extremely high mortality [40]. This experimental design has been successfully applied to several Siberian *L. dispar* populations, where starvation was used as a trigger for covert viral infection [41]. In the current study, the offspring were significantly susceptible to starvation. We believe that this low tolerance to starvation is related to the high prevalence of microsporidia, which, as mentioned above, is not often seen in Siberian *L. dispar* populations.

Contrary to our expectations, the parental insect batches employed in this study proved to be severely affected by microsporidia. On the one hand, the presence of the microsporidia in the parental *L. dispar* generation corrupts the entire experimental design, requires caution in the interpretation of the data, and prevents the extrapolation of the results to interactions between the cypovirus and microsporidia-free populations of the insect host. On the other hand, our data provide an excellent opportunity to investigate an interplay between different stress factors (starvation and viral and microsporidian infections) when applied simultaneously or sequentially to the pest. Our data indicate that microsporidia may be occasionally found in local spongy moth populations, and can cause up to 100% egg infection. This infection, however, may remain unnoticed if the pathogen stays “silent”, unless the insect host is subsequently challenged with additional stress factors.

In this study, it was not possible to confirm the vertical transmission of DsCPV-1 via the spongy moth, an alternative host. Nevertheless, our study clearly answers the main question: DsCPV-1 infection significantly affects the viability of the next generation of *L. dispar*, and this viability is strongly dependent on the magnitude of the pathogen challenge experienced by the parents. Offspring from infected *L. dispar* parents were notably less resistant to temporary starvation, which is typical at high population density of folivorous pests. In the context of the potential application of DsCPV-1 in the field, this finding means that even individuals that survive the viral challenge will produce weak offspring. This is one more benefit of DsCPV-1 for the practical prospects of the development of a DsCPV-1-based biological agent. Of note, the effects of DsCPV-1 did not promote vertical transmission. This may involve some epigenetic mechanisms. Our accidental discovery of the interaction between DsCPV-1 and chronic *V. lymantriae* infection widens the prospects of future research. It seems that there is a mutually reinforcing interaction between microsporidia and DsCPV-1. The results of the current study have important implications for our understanding of how a cypovirus can operate in nature, and how it will affect the next generation of a target species if it is exploited as a bioinsecticide.

## Figures and Tables

**Figure 1 insects-14-00917-f001:**
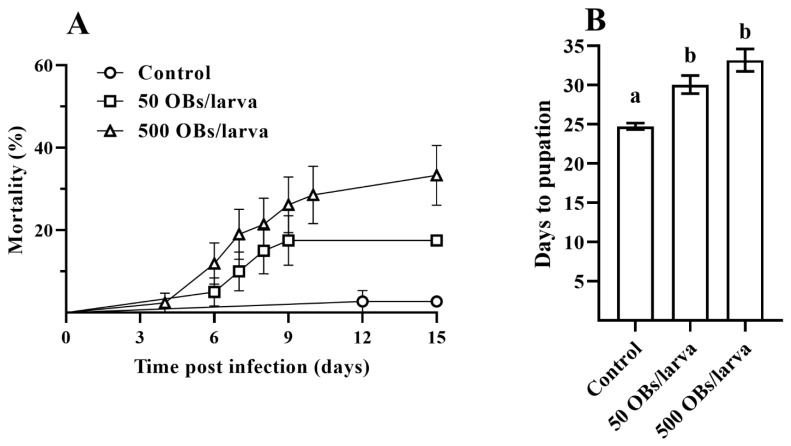
The parental *L. dispar* generation after DsCPV-1 infection. (**A**): mortality dynamics of *L. dispar* larvae (±SE); (**B**): the mean number (±SE) of days to pupation of *L. dispar* larvae. Different letters indicate significant differences (Dunnett’s test, *p* < 0.05).

**Figure 2 insects-14-00917-f002:**
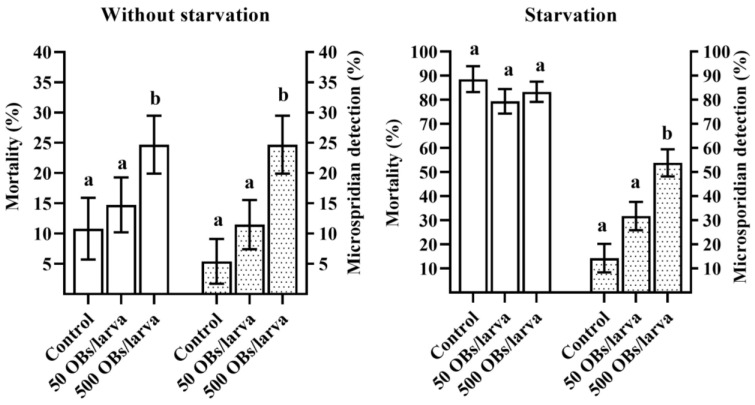
Mortality (±SE, blank columns) induced by starvation and the mean level of detected microsporidia (±SE, dot pattern columns) among starved larval cadavers in the offspring of *L. dispar*. Different letters above the bars indicate significant differences (Dunnett’s test, *p* < 0.05).

**Figure 3 insects-14-00917-f003:**
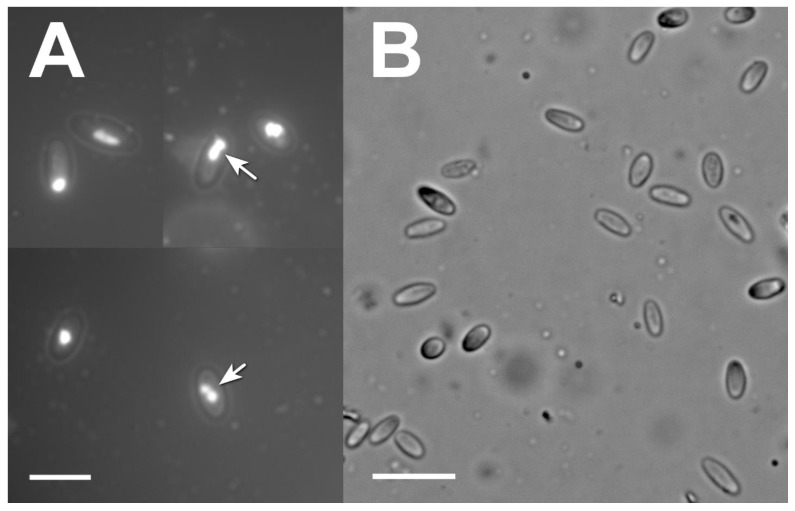
Light microscopy of *Vairimorpha lymantriae* in the offspring of *L. dispar.* (**A**): a combined DAPI-fluorescent and bright-field image of spores containing a distinguishable pair of nuclei (arrows) in a smear preparation of larval cadavers; (**B**): a bright-field image of spores in a squash preparation of eggs. Scale bars = 5 µm (**A**) and 10 µm (**B**).

## Data Availability

The data presented in this study are available on request from the corresponding author.

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
