# Peer review of "The Impact of a Cypovirus on Parental and Filial Generations of Lymantria dispar L."

_insects, 2023, doi:10.3390/insects14120917_

Round 1

Reviewer 1 Report (Previous Reviewer 2)

Comments and Suggestions for Authors

Other aspects to be improved in the manuscript:

Lines 175-176: authors claim LD50 of DsCPV was estimated but the procedure was not described in M&M, in results did not appear such parameters, but this is mentioned again in the discussion.

Line 184: virulence is not accurate term to refer to mortality.

Line 190-196: the impact on offspring of adults viral challenged was not really well addressed. The authors state that adults test negative by qPCR, and also larvae submitted to starvation. Therefore it is not possible to measure DsCPV_1 effect on neither adults developmental traits (fecundity, and fertility) nor offspring developmental parameters. Only a third factor, microsporidian infections can be seen as casual factor of mortality, but we did not know the origin of this pathogen.

Lines 201-205. The authors did not show evidence of sublethal infection in virus-challenge females to support this differential fertility. That could be accomplished by qPCR in this samples.

Lines 206-2029: The epigraph does not correspond to the content. DsCPV has not been detected in the samples processed and therefore it is not possible to speak of its impact on the offspring. There is no causative effect of the DsCPV on mortality demonstrated by this experiment. The trigger effect of starvation to reactivation of virus was not well focus.

Lines 358-359: The results did not support this conclusion because the offspring from virus challenge insect was not proved to harbor virus, the transgeneral transmission had not been demonstrated. 

Comments on the Quality of English Language

A thorough revision of terminology and semantics is necessary.

Author Response

Please find our detail responses in attachment file (.docx)

Reviewer 2 Report (New Reviewer)

Comments and Suggestions for Authors

Akhanaev et al. report the results of a study on L. dispar responses to a cypovirus DsCPV-1 that has applications in biological pest control. The study would be of interest to readers of Insects.
The manuscript requires improvement before it would be acceptable for publication.
My only main concern was the specificity of the qPCR primers that are targeted at a highly conserved gene (polyhedrin) that could show hybridization to the LdMNPV gene. The authors should clarify this.
The other issues relate to clarity and missing information.
I have written suggestions and numbered points on a scanned copy of the manuscript.
Numbered points (see scanned manuscript)
1. Virus common names are not italicized, even when they include the name of the host species.
2. Is this true? Are you saying that viruses are more frequently used than Bacillus thuringiensis? This seems wrong. Please clarify.
3. Explain why stress is a useful way of activating a latent or covert infection.
4. Did you count the OBs? How?
5. Were larvae starved for a period before being inoculated?
6. Were larvae individualized on plants? (one larva per plant)?
7. Were control larvae included in this bioassay?
8. What were the temperature conditions for the bioassay?
9. For how long did insects remain in the refrigerator?
10. What was the sample size of each treatment in section 2.3 (how many insects and how many egg masses?)
11. Was this study replicated?
12. …as a result of the starvation… (sounds like larvae died due to lack of food)? You mean ….larvae that succumbed to disease following the period of starvation stress…?
13. 100 µL of sample….sample of what? Larva homogenized in water? Or 100 µL of larval tissue?
14. Did you include LdMNPV DNA as a control? How do you know that your primers will NOT amplify LdMNPV sequences, as polyhedrin is a highly conserved gene.
15. resuspended in what? Water? What volume?
16. level of microsporidia? Do you mean the quantity of microsporidia in infected individuals? Or the prevalence of infected insects in the population?  I think you mean prevalence of microsporidia-infected individuals, correct? Please clarify.
17. mean values of experimental groups….values of what? Please clarify what you mean here.
18. Your estimate of the LD50 value should be 1687 OB per larva, You cannot justify showing values to two decimal places as your bioassay was not that precise.
19. Indicate that liquefaction would be an indicator of LdMNPV disease.
20. Indicate the prevalence of larval mortality to a percentage point (17 – 33%) or to one decimal place if you have hundreds of larvae in you sample size (17.5 – 33.6%). You cannot justify showing values to two decimal places unless your sample size was in the thousands of insects (which it was not).
21. significantly longer …compared to control larvae.
22. Reword: ….(from the second instar), compared to 30.0 ± 1.1 days and 33.2 ± 1.4 days for insects in the 50 and 500 OB treatments, respectively (Fig 1B….
23. Please show us the data on egg hatch. This could be a new figure of a supplemental figure.
24. Indicate whether control larvae had also been starved.
25. Significantly affected? How?
26. level, meaning? You mean a higher prevalence in the population? Or a higher abundance of microsporidian parasites?
27. Reword to “Possibly” as you have no evidence that the duration of the life stage was the ‘probable’ cause of recovery.
28. Please comment on primer specificity and the possibility of detecting LdMNPV infection in your qPCR assay.
29. You need to explain Table S1 in more detail in the text.
30. Would it be useful to decontaminate food plants with a peroxide or hypochlorite treatment before they are fed to insects in future experiments?

Comments on the Quality of English Language

Requires editing.

Author Response

please find detail responses in attached file (.docx)

Round 2

Reviewer 1 Report (Previous Reviewer 2)

Comments and Suggestions for Authors

no comments

Comments on the Quality of English Language

no comments

Author Response

Thank you, but, we are confused.

Do you have any additional comments or suggestions? 
Because we can see only "no comments".

Reviewer 2 Report (New Reviewer)

Comments and Suggestions for Authors

The authors have addressed my concerns. The manuscript is suitable for publication.

Comments on the Quality of English Language

Requires English editing before it can be published.

Author Response

Thank you

This manuscript is a resubmission of an earlier submission. The following is a list of the peer review reports and author responses from that submission.

Round 1

Reviewer 1 Report

Comments and Suggestions for Authors

This is a rather limited study on the effect of cypovirus infection on L dispar.  The authors have looked to see if virus is transmitted vertically, but have failed to demonstrate this phenomenon, which is common in other insect virus infections. Rather, they have identified the presence of microsporidia infection in insect after infection with the virus.  This was an unexpected outcome and suggests that the microsporidia are a persistent infection of the insects.

The paper only has two figures. The authors could make more of their data if the results described in section 3.1 were turned into a figure.  Visual representation of data is always easier to appreciate than numbers in text format.  

The analysis of virus infection in insects is also rather crude.  Failure to see OBs is not a very sensitive way to assess infection.  I know that in persistently infected insects that harbor baculoviruses you never see OBs.  Rather, PCR is needed to identify virus infection.  Therefore, in this study, the authors should conduct similar PCR-based analysis of insects to identify low levels of virus infection.

This report describes a long term study of virus infection in insects and this reviewer appreciates the time and effort involved.  It would be a shame if the work cannot be published, but does need some additional data to strengthen the story.

Comments on the Quality of English Language

None

Reviewer 2 Report

Comments and Suggestions for Authors

General comment: The results indicate sublethal effect on CPV infected individuals such as larval development time expand and fertility of survival adults. However, no causative effect could be determined from the experimental design related to transgenerational transmission. There is no evidence of vertical transmission of CPV resulted from experiments since viral detection rely on the use of light microscopy for OBs observation and non-specific symptoms found in viral challenged insects.  It would be necessary the use of molecular techniques for specific viral detection that support the presented sublethal effects of virus on host development and viral transmission. On the other hand, starvation is evaluated as a trigger factor of viral activation. Apparently, this factor had no effect on viral activation, but we did not know whether or not the offspring harbour the CPV anyway.

A number of suggestion to authors are made bellow to be taken into account for a final publication.

-        To adapt the tiltle to the manuscript content.

-        To support biological result with the use of PCR for viral detection.

-        To verify CPV transgenerational transmission through viral challenged insect to  offspring.

-        Line 193: to change “infective load” by “infective dose”

-        Discussion. Authors claim that microsporidian infection are transmitted vertically from viral- challenged virus larvae to the offspring, and they suggest there is an effect of CPV dose. It is possible that the inoculum was contaminated with the microsporidian. Have you check for the presence of microsporidia in the viral incoculum? I so, this information should be suministrated as part of the discussion.

Comments on the Quality of English Language

Revision for minor editing is recomended

Reviewer 3 Report

Comments and Suggestions for Authors

In this manuscript, the authors evaluated the effects of infection of Lymantria dispar by DsCPV1, which were previously found by authors to infect L. dispar, on the next generation of L. dispar. They did not confirm that DsCPV1 was not transmitted vertically to the next generation, but found that DsCPV1 infection siginificantly increased the susceptibility of next generation of L. dispar to stress factors and the mortarity caused by Vairimorpha lymantriae.

 The data presented in this manuscript are of great interest to the readership of the journal, however, some points should be addressed prior to publication.

1. In sections 3.1 and 3.2 of the results, it would be helpful to the reader if graphs were also presented for results such as larval survival rates.

2. The names of nucleopolyhedrovirus should be given “Insect scientific name (not italic) nucleopolyhedrovirus. 

Lines 74-75: “Lymantria dispar multiple nucleopolyhedroviruses (LdMNPV)”

 should be written as “Lymantria dispar multiple nucleopolyhedroviruses (LdMNPV)”.